# New Technologies in Knee Arthroplasty: Current Concepts

**DOI:** 10.3390/jcm10010047

**Published:** 2020-12-25

**Authors:** Cécile Batailler, John Swan, Elliot Sappey Marinier, Elvire Servien, Sébastien Lustig

**Affiliations:** 1Department of Orthopedic Surgery and Sport Medicine, Croix-Rousse Hospital, 69004 Lyon, France; jdswan911@gmail.com (J.S.); elliot.sappey-marinier@chu-lyon.fr (E.S.M.); elvire.servien@chu-lyon.fr (E.S.); sebastien.lustig@gmail.com (S.L.); 2Service de Chirurgie Orthopédique, Université de Lyon, Université Claude Bernard Lyon 1, LBMC UMR_T9406, 69003 Lyon, France; 3Cécile BATAILLER, Hôpital de la Croix-Rousse, 103 Grande Rue de la Croix-Rousse, 69004 Lyon, France; 4EA 7424, Interuniversity Laboratory of Human Movement Science, Université Lyon 1, 69100 Villeurbanne, France

**Keywords:** knee arthroplasty, new technologies, patient-specific instrumentation, customized implants, sensors, accelerometers, robotic-assisted surgery

## Abstract

Total knee arthroplasty (TKA) is an effective treatment for severe osteoarthritis. Despite good survival rates, up to 20% of TKA patients remain dissatisfied. Recently, promising new technologies have been developed in knee arthroplasty, and could improve the functional outcomes. The aim of this paper was to present some new technologies in TKA, their current concepts, their advantages, and limitations. The patient-specific instrumentations can allow an improvement of implant positioning and limb alignment, but no difference is found for functional outcomes. The customized implants are conceived to reproduce the native knee anatomy and to reproduce its biomechanics. The sensors have to aim to give objective data on ligaments balancing during TKA. Few studies are published on the results at mid-term of these two devices currently. The accelerometers are smart tools developed to improve the TKA alignment. Their benefits remain yet controversial. The robotic-assisted systems allow an accurate and reproducible bone preparation due to a robotic interface, with a 3D surgical planning, based on preoperative 3D imaging or not. This promising system, nevertheless, has some limits. The new technologies in TKA are very attractive and have constantly evolved. Nevertheless, some limitations persist and could be improved by artificial intelligence and predictive modeling.

## 1. Introduction

Orthopedic surgery is one of the most dynamic medical specialties, with rapid and innovative advances in treatment and surgery. Recently, promising new technologies have been developed in knee arthroplasty.

Total knee arthroplasty (TKA) is a very effective treatment for severe osteoarthritis. TKA leads to good results in returning to daily activities, good survival rates, and overall functional improvement. Satisfying the functional expectations of patients undergoing TKA remains an important surgical goal. Despite advances in surgical technique and postoperative management over the years, up to 20% of TKA patients remain dissatisfied [1]. In a multicenter series of 347 non-selected TKA patients using various implants, only 62% of the patients were totally pain free during gait, 35% were pain free whilst climbing or descending stairs, and 40% complained of pain whilst running [2]. Only 48% of the patients declared being “very satisfied” with the procedure, and 68% considered their operated knee to be “normal for their age.”

Therefore, although TKA is already an effective surgery, the aim is now to further improve patient satisfaction and functional outcomes. Different types of new technologies have been developed to improve surgical accuracy, and as a result, the hope is that there will be improvement in patient satisfaction after TKA. Technologies such as patient-specific instrumentation (PSI), navigation, smart tools, computer or robotic-assisted surgery (CAS) aim to individualize the surgery and account for the anatomy and ligament balancing of each patient, to improve the accuracy of the surgical planning in three dimensions (3D), to increase alignment accuracy and reliability, and improve implant positioning. However, new technologies often have limitations and some disadvantages, and so, knowledge of how to appropriately use these new technologies to improve surgical outcomes is critical.

The aim of this paper was to present some new technologies in TKA, their current concepts, their advantages and limitations, and open a discussion for future use and development.

## 2. Patient Specific Instrumentation

### 2.1. General Concepts

Currently, several orthopedic implant manufacturers offer PSI systems (Smith & Nephew, Wright Medical Technology, DePuy, Biomet, Medacta, and Zimmer). These systems can be used for total knee arthroplasty or for unicompartmental knee arthroplasty. Preoperative 3D imaging (CT scan or MRI) is used to model the knee anatomy and design a personalized surgical plan with respect to bone resection and component positioning and alignment. Once the surgeon approves the plan, cutting blocks or pin guides are rendered and shipped to the hospital, usually in sterile packaging acceptable for the operating room. Pin guides sit on the anterior aspects of the distal femur and proximal tibia to set the placement of pins into the femur and tibia. Whereas custom cutting guides are pinned directly into place and contain cutting slots through which a standard saw is used (Figure 1). These personalized cutting guides allow bone resections to be accurately cut according to preoperative 3D planning.

### 2.2. Results

There are published studies that reported greater accuracy of implant positioning with PSI, however, in different meta analyses, the benefits of PSI on the radiologic results are not clear [3]. Ng et al. reviewed 569 TKAs performed with PSI and 155 with conventional technique using postoperative long-leg radiographs and found significantly less HKA angle outliers (±3°) with PSI than with conventional instrumentation, 9% and 22% respectively [4]. Two meta-analyses investigated the accuracy of alignment. Jiang et al. compiled 18 studies with 2417 patients, demonstrating no significant difference in the number of outliers in mechanical axis as well as coronal, sagittal, and axial alignment [5]. However, Mannan et al., noted favorable femoral rotational outcomes in a meta-analysis of 6 studies on a total of 444 knees [6]. Randelli et al., in a randomized controlled trial on 69 patients reported that PSI did not improve the accuracy of femoral component rotation in TKA in comparison to conventional instrumentation [7]. Other studies described even an unacceptable accuracy with patient-specific cutting block for TKA [8,9].

No study has demonstrated any difference in the clinical or functional outcomes according to the technique (PSI or conventional) [10,11,12,13]. Abdel et al. performed a gait analysis on 40 patients randomized to conventional TKA or PSI and reported no difference in functional or gait parameters after 3 months [10].

Theoretically, the use of PSI should reduce the surgical time, because the operative planning is performed preoperatively, including implant sizing, rotation, femoral and tibial resection. But a recent meta-analysis by Voleti et al. looking at nine studies and 957 patients found a non-statistically significant trend of decreased operative times with a mean of just 5 min per patient [14]. It would be interesting and more relevant to compare surgical time between different new technologies, such as PSI versus CAS.

### 2.3. Advantages and Limitations

The goal of PSI is to plan an accurate cutting guide from 3D imaging and thus aim to improve implant positioning. Frye et al. concluded that an MRI-generated template is better than CT-based guides [15]. MRI is able to account for residual articular cartilage; therefore, the cutting guide can cover a broad contact area and can be directly placed on the bone and residual cartilage of the knee joint. CT is unable to account for residual cartilage; the corresponding cutting guide has to rely on multiple bony sites. The use of preoperative 3D imaging can facilitate the “mini-invasive” surgery, with limited exposure. And there is no violation of intramedullary canals, with decreased risk of fat embolus and blood loss. PSI can also potentially increase surgical efficiency by reducing operative time and cost compared to robotic-assisted surgery.

By contrast, PSI systems do not aid in the performing of gap balancing or soft tissue releases. These steps, which are crucial to surgical success, are performed as usual by the surgeon. Tibial component rotation and implant fixation, as well as patellar preparation, also remain the responsibility of the surgeon.

To complete the benefit of an accurate cutting guide, some authors have associated this technology with another, such as the use of sensors.

## 3. Individual Knee Arthroplasty Implant

### 3.1. General Concepts

Nowadays, surgeons can choose components from a wide range of sizes, including standard width and narrow and sometimes asymmetrical tibia components. However, anatomic variations are not limited to large or narrow, but also include several other features, such as the trapezoidicity of the distal femur, the condylar radii of curvature, joint-line obliquity, and the shape of the trochlea and tibial plateau.

A customized individually made implant device is conceived and designed to reproduce the native (pre-arthritic) anatomy of the knee, using single-use customized instrumentation. Customized implants have been slowly adopted in operating theaters since their introduction around 2011. The main aims are to optimize bone implant fit and avoid prosthetic overhang or under-coverage, to improve ligament balancing by avoiding resection laxity, to improve mid-flexion stability and kinematics by restoring the native radii of curvature, to improve patellofemoral tracking by restoring the native femoral torsion and customized trochlea, and to facilitate the restoration of the native limb alignment.

The design and manufacturing process of the custom TKA takes 6–8 weeks and requires cooperation between the surgeon and engineers. A 3D model, which is made by converting a series of 2D scanned images of the knee joint, is used to fabricate a customized implant and instrumentation by using additive manufacturing/3D printing technologies. Operative planning and the implant design are always validated by the surgeon. The surgery is performed with personalized cutting blocks for the femur and the tibia. All surgical parameters such as implant size and positioning and limb alignment are decided preoperatively.

The customized implants are frequently compared with the “off-the-shelf” implants, which are fabricated in a range of standard sizes. Surgery performed using “off-the-shelf” aims matches the most appropriate implants for each patient and to aim for good functional results with lower cost.

### 3.2. Results

Currently, few studies have reported the results of customized implants for TKA. Interesting results have been described in implant positioning and limb alignment [16,17,18,19]. In a cohort of 258 custom TKAs, Bonnin et al. have reported that 84% of TKA had satisfactory mechanical femoro-tibial axis (HKA), mechanical femoral axis (FMA), and mechanical tibial axis (TMA) [16]. Deviation between the planned and postoperative angles was −0.5° ± 1.8° for FMA, −0.5° ± 1.8° for TMA, and −1.1° ± 2.1° for HKA angle. Arbab et al. described, in a case-controlled study, that patient-specific TKA demonstrated fewer outliers from neutral coronal leg alignment compared to conventional technique (16% in the patient-specific TKA cohort and 26% in the conventional TKA group) [17].

Very few studies have reported clinical outcomes after custom implants. Reimann et al. described that the patient-specific implants might increase patient satisfaction at 2 years [20]. Schwarzkopf et al. and White et al., found no significant difference in clinical outcomes, with only a tendency to decreased range of motion [21,22]. More studies are needed to assess the functional outcomes after patient-specific implants.

### 3.3. Advantages and Limitations

This technology is based on the theory that many unsatisfactory outcomes or residual pain after TKA can be attributed to a lack of anatomic restoration that can be difficult to identify by clinical examination. Chronic pain, stiffness, or laxity can be secondary to incorrect sizing or malrotation of the femoral implant [23,24]. A customized implant is thus designed to reproduce the native anatomy. Customization of the bone cuts and implants also allows engineers to minimize, as much as possible, the thickness and weight of the implants and the quantity of bone resection. Ligament balancing also seems to be simpler for the surgeon when using customized implants, particularly at mid-flexion, due to the conservation of the condylar curvature radii.

Nevertheless, the use of customized implants does not take into account ligament balancing, and so ligament balancing remains dependent on the surgeon’s intraoperative assessment. Bonnin et al. described 46% of tibial bone recuts in 258 custom TKAs [16]. Moreover, the preoperative planning is performed using imaging of an arthritic knee to restore a knee to pre-arthritic function. In knees with severe deformity or advanced stage osteoarthritis, the final outcome could still be unsatisfactory with no guarantee of superior results.

## 4. Sensors in TKA

### 4.1. General Concepts

The current advances in knee arthroplasty result in improved accuracy in implant positioning, limb alignment, implants sizing, and reduced soft tissue damage. However, one of the remaining challenges in achieving a satisfactory TKA is ligament balancing. Ligament balancing is essential during TKA and especially difficult to assess and to manage. Experienced surgeons traditionally obtain soft tissue balance using their own subjective “feeling” rather than a scientific perspective [25,26,27,28]. The ligament balancing “feeling” is affected by many factors such as patient obesity, gender, generalized laxity, degree of joint contracture, surgical experience, and even the surgeon’s daily condition [29,30]. Poor ligament balancing can cause instability, stiffness, pain and TKA revision or patient dissatisfaction [31]. For example, TKA revision for instability has been estimated as greater than 20% each year [32].

New technologies can assess ligament balancing, such as a robotic-assisted system which can register the preoperative laxity and compare this to the planning of the TKA. Navigation also takes into account preoperative laxity and its evolution after the bone resections are made and the implants are positioned. Nevertheless, for each case (robotic or navigation), the laxity assessment at the beginning of the surgery is manual. The surgeon exerts a significant varus and valgus force on the knee. This assessment is dependent on the strength of the surgeon, on the depth of anesthesia, and the BMI or the physical stature of the patient. The ligament balancing utilizing these advanced technologies is not entirely objective and remains challenging and dependent upon the surgeon’s experience with the system. The PSI, the customized implants, and the accelerometers are useful tools to determine the bone cut axis and implant positioning but are completely independent of the ligament balancing.

The purpose of sensors is to give objective data on soft tissue balancing during TKA. These disposable devices deliver wireless data to an intra-operative monitor to facilitate informed decision-making regarding implant position and soft tissue releases to improve balance and stability through a full range of motion. Different manufacturers produce this device, but it is mainly VERASENSE Knee System (OrthoSensor Inc., Dania Beach, FL, USA). This system is a wireless and disposable articular loading quantification device, which is inserted in the tibial component tray during the surgery, after the tibial and femoral cuts are completed (Figure 2). The capsule is closed by few stitches. The surgeon holds the leg in a neutral position and monitors the medial and lateral loading forces from full extension to full flexion. Less than differential loading of 15 pounds between the medial and lateral compartments, is considered as adequately “balanced” [33]. After initial ligament balance assessment, if the joint shows imbalance, additional soft tissue releases or bony resection can be performed.

### 4.2. Results

Several studies have described and assessed the results and the consequences of the sensors on ligament balancing of a TKA. Cho et al. have found that an objective quantification using real-time orthosensor improved the soft tissue balance in TKA (for measured resection TKA or modified gap balancing TKA) [34]. In a prospective cohort of 50 sensor-assisted TKAs (without severe deformity), Song et al. reported that 74% of knees needed an additional rebalancing procedure with the sensor, after conventional gap balancing with the tensiometer [35]. There were coronal and sagittal load imbalances in the evaluation using the sensor, even after the achievement of an appropriate gap balance using the tensiometer. However, few studies have reported an improvement of the functional outcomes after sensor-assisted TKA compared to conventional TKA, and several limits are often present. Chow and Breslauer reported that the clinical scores and range of motion (ROM) were significantly higher after sensor-assisted TKA than after manually balanced TKA, but without preoperative radiographic evaluation [36]. Geller et al. reported that the use of the sensor significantly reduced the rate of arthrofibrosis [37]. Song et al., in a comparative study of 50 sensor-assisted TKAs, described no clinical and radiological difference between TKA with or without using a sensor system [38]. The clinical follow-up remains too short to assess the clinical benefit of this device.

### 4.3. Advantages and Limitations

Apart from the cost, the principle limitation in the assessment of the sensors is that the normal range of joint compartment pressures is not well understood. What are the pressure values in a well-balanced TKA? Several studies have demonstrated than soft tissue imbalance is a major cause of dissatisfaction, and the subjective assessment of surgeons is insufficient to ensure an appropriate soft tissue balancing [39,40]. Indeed, some systems can produce an equal and rectangular extension and flexion gaps, such as tensiometers or navigation systems [41,42]. However, no study has demonstrated a significant improvement of patient satisfaction after TKA with these devices [43,44,45]. Nagai et al. reported that the medial compartment was always stiffer than the lateral structure at all flexion angles from 0° to 135° [46]. Nevertheless, the appropriate loads are defined mainly with arbitrary limits. Meneghini et al. reported that the Knee Society Objective Score remained favorable under high medial compartment loads (>75 lbs; >34.0 kg) but decreased significantly under high lateral compartment loads (>75 lbs; >34.0 kg) [47]. The “normal” values remain extended, and a personalized adaptation is probably required for each patient.

## 5. Accelerometer

### 5.1. General Concepts of an Accelerometer

Accelerometers are smart tools developed to improve the alignment of femoral and tibial components in TKA. Proper component alignment is important to improve the likelihood of functional restoration, patient satisfaction, and TKA survivorship [48,49]. Currently, the ideal alignment in TKA remains controversial. Nevertheless, despite the type of alignment (mechanical, kinematic, restricted kinematic), the accuracy of the tibial and femoral cuts remains important. This is especially important for kinematic alignment because an error of 3° in the component alignment has serious consequences when the targeted alignment is already in varus or valgus. That is why these tools are useful particularly during surgeries with individualized component alignment. In a previous study, the recommended alignment in the coronal plane (within 3° of a neutral mechanical axis) was achieved in only 70–80% of patients undergoing conventional TKA using extra and intramedullary guides [50].

Accelerometer-based navigation is a portable surgical navigation system that does not use a large computer console for TKA (Figure 3). The first validation study about this system was in 2011 (cadaveric study) [51]. Accelerometer-based navigation is a handheld, sterile device used to determine the resection planes of the distal femur and the proximal tibia in the coronal and sagittal planes [52]. These systems are wireless and imageless, and they capture data during the procedure, and directly display the data on pods, which are attached to the femoral and tibial resection guides within the surgical field. For the distal femoral resection, the surgeon impacts a spike in the distal femur on the distal point of the femoral mechanical axis. An electronic pod is attached to this spike. The center of the hip is registered through a “stop-and-go” movement with star configuration, and femoral mechanical axis is detected. The distal femoral cutting block is then attached to the pod guide, and the resection is adjusted in coronal (varus/valgus) and sagittal (flexion and extension) planes referring to mechanical axis, according to the wished planning. After the cut is performed, a validation pod can be used to confirm the cut axis and perform additional resection if necessary. The other femoral cuts are performed as usual with a conventional resection guide. For the proximal tibial resection, a new spike is impacted on the tibial spines. An electronic pod is attached to an extramedullary guide, also attached to the spike. Another electronic pod is attached to the ankle. A registration of the tibial mechanical axis is performed with knee movements from left to right then with small movements of knee flexion. The tibial resection guide is inserted on the extramedullary guide and is then adjusted to correct the coronal alignment and the slope according to the wished planning. After resection, the accuracy of alignment can be validated by an electronic pod, and any adjustments can be performed if needed.

The trials and the implants are positioned as usual with a conventional technique. The femoral and tibial rotations are determined manually as in the conventional technique.

### 5.2. Results of Accelerometers

Budhiparama et al. have described the results of the main studies about accelerometers in a systematic review [53]. Five of nine studies favored accelerometer-based navigation for the restoration of HKA [54,55,56,57,58], while four of nine studies found no differences between the study groups [59,60,61,62]. In terms of coronal-axis alignment of the femoral component, seven of nine studies favored accelerometer-based navigation [54,55,56,57,58,62]. There was no significant difference in sagittal alignment.

Only two of six studies that evaluated the proportion of patients with outlier alignments supported accelerometer-based navigation [57,58]; the other four found no differences in this important endpoint [54,55,60,61]. Li et al. have reported similar results in a meta-analysis on 275 total knee arthroplasties performed with the iASSIST navigation system [63]. This system provided significantly increased accuracy in the coronal femoral angle (*p* < 0.00001) and the coronal tibial angle (*p* < 0.00001) compared with conventional techniques. There was no significant difference in functional knee score at short term follow-up in the iASSIST group compared with the conventional group [63]. No studies have demonstrated better clinical outcomes with accelerometer-based navigation compared with conventional technique TKA [53].

Most studies did not describe longer surgical times with the accelerometer-based navigation compared with conventional techniques [54,55,57,61,62]. No learning curve has been assessed in the literature, likely because this is only a tool and not a complete navigation system. There were no differences concerning the complication rate between these two techniques. No reoperations or revisions have been described following the use of accelerometer-based navigation.

### 5.3. Advantages and Limitations

Budhiparama et al. reported that they “found very inconsistent (and generally small) benefits in favor of accelerometer-based navigation in terms of alignment, but no benefits regarding the functional outcomes or the risk of complications or reoperations. Until or unless more compelling evidence in favor of the new technology emerges, they recommend against its widespread adoption” [53].

The real benefit of this accelerometer-based navigation is difficult to prove because the aim is to decrease the alignment outliers. The consequences on the functional outcomes are thus less clear [55]. It is used as an adjunct tool during TKA and the clinical follow-up of these studies is very short. The assessment of the long-term revision rate according to the surgical technique is not currently possible, and there are very few high levels of evidence studies assessing this accelerometer-based navigation.

This system has limitations. The axis of the bone cut is dependent on the reference points chosen on the patient. Thus, if the knee center is incorrect by some millimeters, the mechanical axis will be incorrect by some degrees. The accuracy of the references is very important and dependent on surgeon precision. Moreover, this system assists only in component alignment and does not assist with component sizing, component rotation, ligament balancing, or even for the target alignment. It can be considered simply as a tool to improve bone cut accuracy compared to extra or intramedullary guide techniques.

Several advantages have been described regarding smart tools in TKA. Conventional CAS systems with large console computers are criticized because they seem to increase operative time and cost, are associated with long learning curves [49,64], have sensitive optical instruments [65], and do not report an improvement in component survival. The accelerometer-based navigation appears as a simpler and potentially less expensive alternative. Moreover, it can be used with almost all TKA. Indeed, it is not dependent on particular types of implants.

Currently, the accelerometer-based navigation is an interesting system for complex cases with extra-articular deformities (post-traumatic or developmental). Several studies described difficult cases with severe deformities and the impossibility of using an intramedullary guide [66,67]. The accelerometer-based navigation is a simple tool to facilitate a TKA with extra-articular deformity.

## 6. Robotic-Assisted Knee Arthroplasty

The prevalence of robotic-assisted surgery is a natural evolution from computer-assisted surgery, which has been used for knee arthroplasties for over 20 years. The main benefit offered by robotics is accurate and reproducible bone preparation due to a robotic interface, whatever system is used [68,69]. This robotic-assisted system also allows an assessment of the ligament balancing according to the bone cuts and the implant positioning during the surgery. This ligament balancing is usually related to the valgus or varus stress given by the surgeon. The aim of robotic systems is not to replace the surgeon, but to improve their performance.

### 6.1. General Principles

#### 6.1.1. Image-Guided Versus Image-Free Surgical Planning

Current robotic systems require the creation of a 3D plan based either pon an intraoperative bone morphology mapping, or a preoperative CT scan. Preoperative CT imaging includes some slices on the ankle and the hip, as well as the knee, to determine the 3D mechanical femorotibial axis. A 3D reconstruction is created to template component size and positioning [70]. The surgical planning is completed preoperatively. During surgery, the robotic arm assists in performing very accurate bone cuts according to the surgical plan. The disadvantages include the cost of the preoperative imaging study, the patient inconvenience to obtain the study at certified centers, and the radiation exposure [70,71,72].

Alternatively, image-free robotic-assisted systems need an intraoperative registration of the anatomical surfaces by a manual bone surface mapping. A 3D virtual model is then created, and the planning is performed during surgery. Preoperative 3D imaging is unnecessary and no specific planning is performed preoperatively. Thus, the intraoperative registration relies on the surgeon’s precision of inputting the correct data points, which is subject to human error.

#### 6.1.2. Autonomous, Semiautonomous, and Passive Robotic Systems

Three categories of robotic systems for knee arthroplasties exist: passive, semiautonomous, and autonomous robotic systems. A passive system provides a 3D virtual model, which allows accurate preoperative planning. But there is no system to prepare the bone. The autonomous and semiautonomous systems incorporate safeguards against the removal of bone beyond the 3D plan (Figure 4).

The passive systems are a computer-assisted or navigation system and perform accurate surgical planning and can guide the tool positioning, but the bone removal is performed only by the surgeon.

With the autonomous robotic-assisted system, the surgeon performs the surgical plan (bone resections, implants positioning, and sizing), the initial approach, and the knee exposure. Then the robotic system has the capability of completing the remaining surgery without surgeon input. Nevertheless, the surgeon can control an emergency switch to stop the procedure or to adjust the plan. CASPAR (Ortho-Maquet/URS, Schwerin, Germany) and ROBODOC (Curexo Technology Corporation, Fremont, CA, USA) are autonomous robotic systems, based on preoperative CT imaging.

The semiautonomous robotic-assisted systems combine the benefits of a navigation system and of an autonomous robotic system. The surgical planning is performed by the surgeon, either based on preoperative 3D imaging or on intraoperative bone surface mapping. Semiautonomous robots are controlled and manipulated by the surgeon. But the surgeon’s control is modulated by the robot to limit bone preparation to the surgical plan. Thanks to a feedback loop, the bone removal is controlled (with a saw or a burr), or the cutting guide is positioned. The surgeon cannot deviate from the planned bone resection. This control improves the surgeon’s accuracy and decreases the risk of errors. These systems include the image-free robotic Navio system (Smith & Nephew, Memphis, TN, USA), the image-based MAKO robotic arm (Stryker, Mahwah, NJ, USA), the ROSA knee system (Zimmer Biomet, Warsaw, IN, USA) and OMNIBOT (OMNIlife Science, Inc.; Raynham, MA, USA).

#### 6.1.3. Ligament Balancing

Ligament balancing during knee arthroplasty is critical to obtain good functional outcomes and maintain normal knee kinematics. These systems can register the ligament balance or imbalance before the intervention, the planned ligament balancing, and the balance at the end of the procedure. During all of the steps of the surgery, the surgeon can assess the ligament balancing can make adjustments. Depending on the robotic-assisted system, the ligament balancing can be assessed in extension and knee flexion at 90°, or during all range of motion. Current robotic systems incorporate soft tissue balancing algorithms in their planning and procedures for unicompartmental knee arthroplasty (UKA) and TKA. This system is very helpful, particularly for mid-flexion balancing, which is very difficult to assess during conventional surgery. Nevertheless, the assessment of the ligament balancing is always manual. The robotic system registers only the varus and valgus stress exerted by the surgeon. These values are not entirely objective and again depend on the surgeon and multiple patient and intraoperative factors.

### 6.2. Results

#### 6.2.1. UKA

With image-based and image-free robotic-assisted systems, the results and particularly the positioning of the implants have been significantly improved [73,74,75,76,77,78]. Herry et al. have shown that the joint line level was very well controlled with the robotic system [78]. Ponzio and Lonner have also reported that aggressive tibial resection is less frequent during robotic-assisted UKA [79]. In the literature, the mean implant positioning is not always significantly improved with robotic-assisted UKA. But, the reduction of outliers is significant [73,74] and thus relevant to the reduction of failure. Several meta analyses and systematic reviews reported similar results [69,80]. Studies have also suggested that a robotic-assisted system leads to improved ligament balancing [81], decreased post-operative pain [82], faster return to work [83] and to sport [84], and to better functional scores [82].

Studies on robotic-assisted UKA report satisfying short and medium-term survival rates [85,86]. Nevertheless, no comparative study has demonstrated a better survival rate for robotic-assisted UKA, compared to conventional UKA. Published rates of revision after robotic-assisted UKA vary from 3% to 10% at mid-term [87,88,89].

By contrast, there is less evidence for improvement in functional outcomes after robotic-assisted UKA compared with the conventional technique [86]. Gilmour et al. reported that more active patients may benefit from image-based robotic-assisted surgery [86].

#### 6.2.2. TKA Results

There are very few studies on robotic-assisted TKA. These studies are essentially preliminary studies without a comparative group or long-term follow up [68].

Only the autonomous robotic-assisted system currently has a long follow-up. Using the Robodoc and a minimum follow-up of 10 years, Kim et al. found no differences between robotic-assisted TKA and conventional TKA in terms of functional outcome scores, aseptic loosening, overall survivorship, and complications [90].

Image-based robotic-arm assisted TKA (Mako system) improved accuracy in achieving the planned implant position compared to conventional TKA [91,92]. Kayani et al. reported that the image-based robotic system improved the accuracy of femoral sagittal and coronal alignment, tibial sagittal and coronal alignment, tibial slope and limb alignment, and joint line restoration, compared to a conventional technique [91]. The rate of early complications was not significantly different between robotic-assisted TKA and conventional TKA [93,94].

Kayani et al. reported that image-based robotic-assisted TKA was associated with reduced bone and periarticular soft tissue injury compared with conventional TKA [95]. Several case-controlled studies assessed the short-term functional postoperative scores, with a maximum follow-up of 17 months, between robotic-assisted and conventional TKA, with inconclusive results [96,97,98,99]. More investigations at mid and long-term are necessary for the semiautonomous robotic-assisted system. The new technologies, and particularly the robotic-assisted surgery, are also very interesting tools for the controversial question of the limb alignment. Indeed, these systems allow for accuracy in implant positioning and limb alignment and can facilitate the alignment reflection. Some authors have even described an adaptation of the kinematic alignment with the robotic-assisted system [100,101].

### 6.3. Recent Developments

#### 6.3.1. Patellofemoral Arthroplasty

There are different considerations when performing a patellofemoral arthroplasty. However, it is probably one of the best indications for robotic surgery since the ideal position can only be obtained through accurate mapping of the 3D anatomy of the distal femur. The 3D planning stage is now much easier thanks to the robotic surgical system which produces a 3D model of the trochlea and records landmark points (medial and lateral epicondyle points, Whiteside’s Line, femoral mechanical axis, and mechanical femoral valgus). It displays a 3D simulation of the trochlear component placement and guarantees a perfect transition area between the femoral component and femoral condyle cartilage prior to the bone cut. The preparation stage is made easier thanks to the controlled bone removal using a robotic handpiece that removes the residual cartilage and subchondral bone in line with the planning, obtaining a more consistent outcome than standard tools.

Very few studies described the results of robotic-assisted patellofemoral prostheses, and these studies are not comparative. The first series reported satisfying functional scores and good implant positioning [102].

#### 6.3.2. Bicruciate-Retaining Arthroplasty

There is a long history of bicruciate-retaining total knee replacements with promising long-term results but a reputation as a technically demanding procedure. Robotics have provided considerable assistance for surgeons undertaking this type of arthroplasty, which requires a meticulous understanding of the difference in joint space between the lateral and medial compartments, as well as highly accurate bone preparation. Protecting the tibial spines is also much easier when using a bur guided by a robotic handpiece.

#### 6.3.3. Combined UKA and Anterior Cruciate Ligament (ACL) Reconstruction

ACL reconstruction combined with unicompartmental knee arthroplasty is a tempting solution for anyone keen on unicompartmental procedures. The robotic system ensures accurate implant positioning (with tibial slope and overall alignment control in particular) but also allows the surgeon to visualize any residual gap before and after implant fixation. Both implant position and the polyethylene thickness can be adjusted based on the dynamic data provided by the robotic system.

#### 6.3.4. Bicompartmental Arthroplasty

In some cases, if the patient is young and active with bicompartmental osteoarthritis, there may be an indication for two partial knee replacements (usually a medial unicompartmental arthroplasty and a patellofemoral replacement) (Figure 5). Despite long-standing support for this surgery, in particular from Philippe Cartier, it is technically challenging. The Navio^®^ system can be used to predict and adjust the relative position of the two implants, making this uncommon procedure more consistent.

### 6.4. Learning Curve and Specific Complications

The new technologies and new systems need an assessment and particularly an evaluation of the learning curve. In fact, learning curves for robotic assistance in knee replacement surgery have been demonstrated to be quite short [91,103]. Kayani et al. showed that robotic-assisted UKA is associated with a learning curve of seven cases for operative time [104]. Lonner et al. [105] did a retrospective study of 1064 UKA performed with either the Navio system (Smith and Nephew^®^) or the MAKO system (Stryker^®^) and reported no soft tissue or bone injuries or complications related to the use of robotic-assistance. Blyth et al. [5] in study on UKA performed with the assistance of the MAKO system found no complications. Only some minor complications related to the use of navigation pins are described [105,106]. Similar results have been described with the Navio system [107].

Sometimes, the use of the robotic system was aborted during the procedure. The rate of intra-operative switching from the robotic to a conventional technique varies in the literature (1–22%) [108,109,110]. These occasions all occurred in the learning curve of the robotic-assisted system. A good knowledge of this robotic system by the surgeon and the nurses is critical to avoid errors during this procedure.

## 7. Conclusions

The new technologies in TKA are very attractive and have constantly evolved. They expand the possibilities and the surgical indications, improve our knowledge of knee biomechanics, and try to restore native knee function. Nevertheless, all technologies need long-term assessment and critical appraisal. The limitations of new technologies could be improved by artificial intelligence and predictive modeling. Many medical technology and health insurance companies, as well as arthroplasty registries are already collecting data on patient demographics, implant survivorship, causes for revision, and patient-reported outcome measures. Creating accurate, reproducible, and predictive algorithms may one day provide advanced tools for shared decision making regarding surgical indications and predicting expected outcomes in knee arthroplasty.

## Figures and Tables

**Figure 1 jcm-10-00047-f001:**
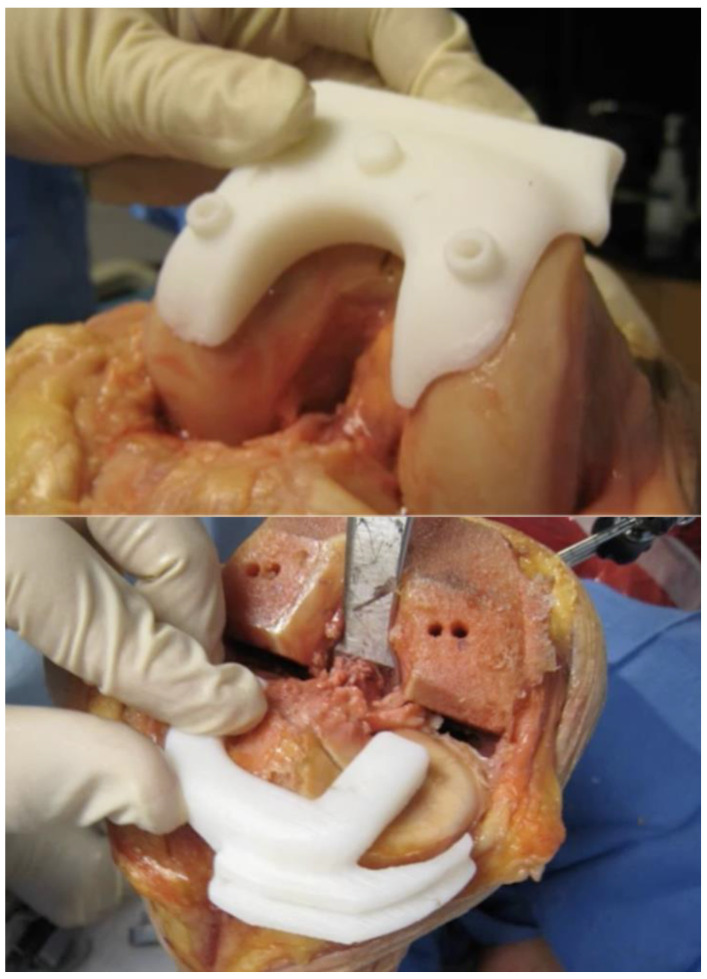
In the patient-specific instrumentation, the personalized cutting guides can be pinned directly into place in the femur and the tibia and contain cutting slots for a saw.

**Figure 2 jcm-10-00047-f002:**
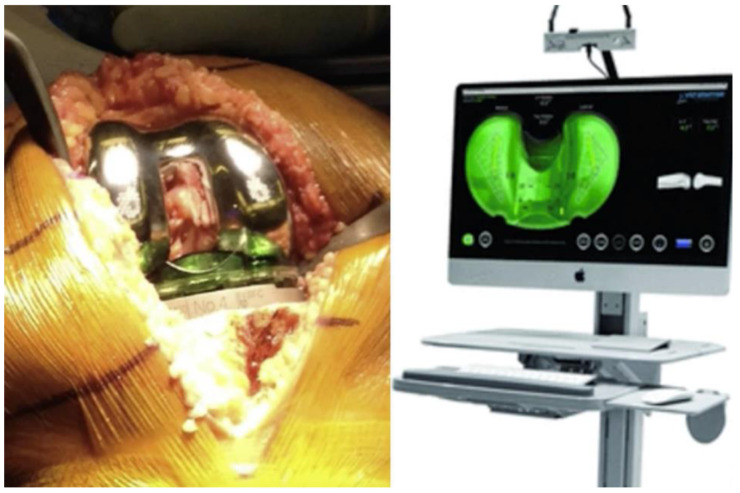
The sensor is an articular loading quantification device, which is inserted in the tibial component tray during the surgery, after the tibial and femoral cuts are completed (Orthosensor, Verasense).

**Figure 3 jcm-10-00047-f003:**
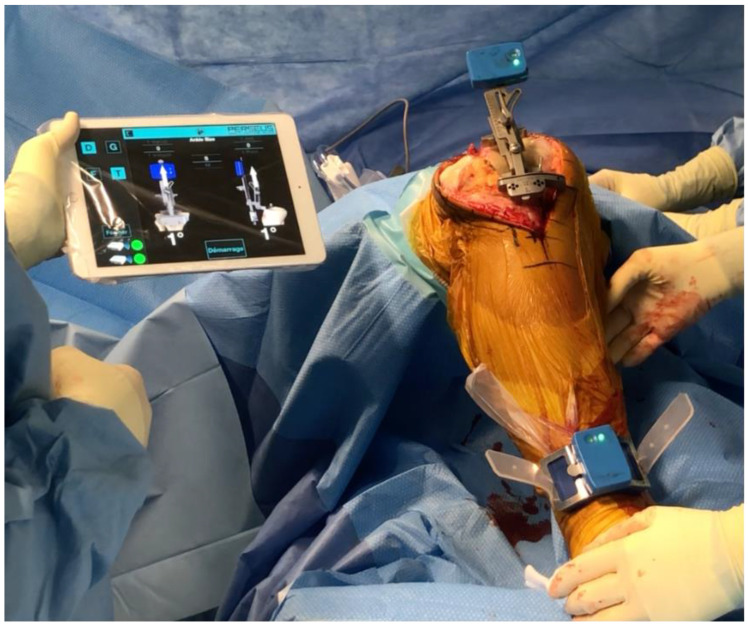
The accelerometer is a handheld device used within the operative field to determine the resection planes of the proximal tibia. This system guides resection angles in the coronal and sagittal planes (Perseus, Orthokey).

**Figure 4 jcm-10-00047-f004:**
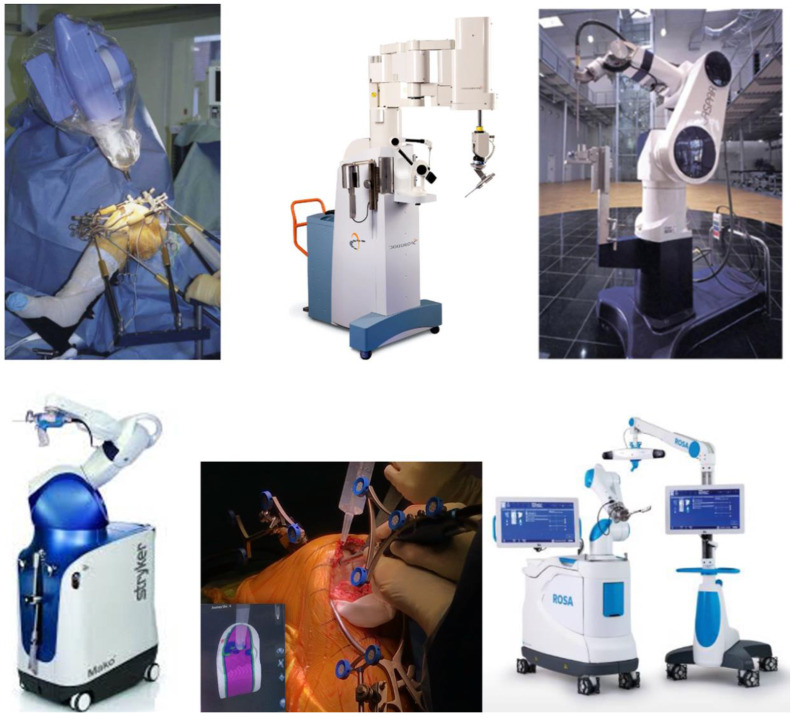
The autonomous and semiautonomous robotic systems incorporate safeguards against removal of bone beyond the 3D plan. The first three robotic systems are autonomous (Robodoc, Caspar). The last three robotic systems are semiautonomous (Mako, Navio, Rosa).

**Figure 5 jcm-10-00047-f005:**
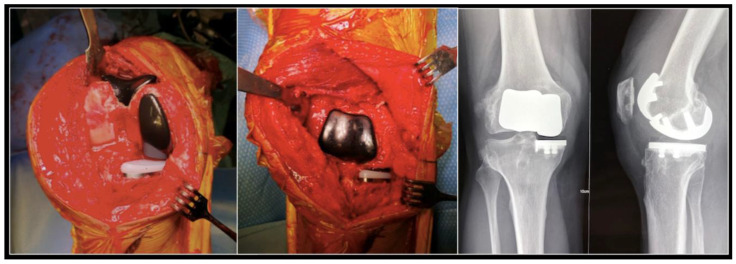
Per operative pictures and radiographs of a medial unicompartmental arthroplasty associated to a patellofemoral replacement in a young and active patient.

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
