# Peer review of "New Technologies in Knee Arthroplasty: Current Concepts"

_jcm, 2020, doi:10.3390/jcm10010047_

Round 1
Reviewer 1 Report
The Study needs to be restructured since there are no hypothesis stated
Please first stae a hypothesis, e.g. higher accuracy leads to better results in TKA....
and then structure the article
Author Response
Dear Reviewer,
Thank you to have reviewed our paper. We are completely agreed with you. The structure of the paper is not appropriate for a systematic review. But this paper by no means is intended to be a systematic review to answer to a hypothesis. The aim was to bring some current concepts on new technologies in knee arthroplasties. We can’t be exhaustive in a systematic review about all new technologies in knee arthroplasties in only one paper.
That’s why we have decided to present this work as current concepts without hypothesis.
Maybe the paper is not submitted in the appropriate section as suggested by the second reviewer (in the form of an editorial?).

Reviewer 2 Report
I would like to thank the author for allowing me to review their work.
This is an interesting narrative.
The author present the current technologies in knee replacement.
However, this does read more like an editorial than review.
This is not a systematic review with no search strategy in a methods section.
I would have liked to see explained how the new technologies discussed have allowed concepts such as kinematic alignment gain in popularity.
I would be happy to support publication in the form of an editorial if this is in line with the Journals publication policy.
Author Response
Dear reviewer,
Point 1: However, this does read more like an editorial than review.
This is not a systematic review with no search strategy in a methods section.
Response 1:
It is the same answer than for the first reviewer.
“Thank you to have reviewed our paper. We are completely agreed with you. The structure of the paper is not appropriate for a systematic review. But this paper by no means is intended to be a systematic review to answer to a hypothesis. The aim was to bring some current concepts on new technologies in knee arthroplasties. We can’t be exhaustive in a systematic review about all new technologies in knee arthroplasties in only one paper.
That’s why we have decided to present this work as current concepts without hypothesis.”
Point 2: I would have liked to see explained how the new technologies discussed have allowed concepts such as kinematic alignment gain in popularity.
Response 2:
Thank you for your comment. Indeed, these new technologies bring interesting reflections about some concept, particularly the limb alignment. We have added this aspect in the manuscript. L406-409 P 11
Point 3: I would be happy to support publication in the form of an editorial if this is in line with the Journals publication policy.
Response 3:
Thank you to support this publication. We agree with you it is not a systematic review but current concepts. We leave it up to the editorial to decide which journal section is the most appropriate.

Reviewer 3 Report
The aim of this paper was to present some new technologies in TKA, their current concepts, their advantages and limitations.
The authors discussed the technique of PSI and the use of customised cutting guides, and described no difference in the outcome with regard to clinincs and function. This section is well summarized, with some of the important studies and metaanalyses included.
Next, the technique of individual implants was introduced. Here, although there are just a view studies available, the results seem to be promising with regard to a more accurate mechanical axis when compared to the preoperative planning and more satisfaction among the patients.
Quite a long section of the manuscript is related to the use of sensors in TKA. The purpose of sensors is to give objective data on soft tissue balancing during TKA in order to facilitate implant positioning and/or soft tissue releases to improve balance and stability.
Personally, I do not have any experience with this technique, but this section was summarized and written properly.
Recent studies were presented to the reader with reagrd to the outcome and device-specific problems and limitations. The one to mention is the lack of understanding regarding the range of joint compartment pressures.
The next section referred to the use of accelerometers, which guide resection angles in the coronal and sagittal planes in order to confirm alignment accuracy of the components. This technique is not described understandable to the readers in my opinion. The introduction focussed on general complications in alignment restoration during TKA and does not provide any information about the application of the presented tool itself.
Please revise this section accordingly.
The section of robotic assisted techniques in TKA is a very nice summary yet again. The authors described the technique, as well as indications and results of robotic surgery in Terms of axis restoraration, ligament balancing, surgeons learning curve and application (UKA, TKA, patellofemoral arthroplasty, combined techniques) in a very accurate manner and provide current literature.
Overall, I do recommend accepting this manuscript for publication since this is a very nice overview of current concepts in knee arthroplasty.
Thank you very much for your work.
Author Response
Dear Reviewer,
Thank you for your comments.Point 1:The next section referred to the use of accelerometers, which guide resection angles in the coronal and sagittal planes in order to confirm alignment accuracy of the components. This technique is not described understandable to the readers in my opinion. The introduction focussed on general complications in alignment restoration during TKA and does not provide any information about the application of the presented tool itself.
Please revise this section accordingly.
Response 1:
We have revised this section about accelerometers. We have shortened the part about TKA alignment. And we have tried to describe accurately the surgical technique with the accelerometers.§5.1 general concepts L232 P6
P7 L259-275
